# Recovery of Rare Earth Oxide from Waste NiMH Batteries by Simple Wet Chemical Valorization Process

**Nak-Kyoon Ahn [1]** , **Basudev Swain [1],\*** , **Hyun-Woo Shim [1]** and **Dae-Weon Kim [2],\***

1   Materials Science and Chemical Engineering Center, Institute for Advanced Engineering (IAE),
    Yongin 17180, Korea; ahnk@iae.re.kr (N.-K.A.); scode@iae.re.kr (H.-W.S.)
2   Advanced Materials and Processing Center, Institute for Advanced Engineering (IAE), Yongin 17180, Korea
\*   Correspondence: Swain@iae.re.kr (B.S.); mdsimul@iae.re.kr (D.-W.K.)

**Abstract:** Nickel metal hydride (NiMH) batteries contain a significant amount of rare earth metals (REMs) such as Ce, La, and Nd, which are critical to the supply chain. Recovery of these metals from waste NiMH batteries can be a potential secondary resource for REMs. In our current REM recovery process, REM oxide from waste NiMH batteries was recovered by a simple wet chemical valorization process. The process followed the chemical metallurgy process to recover REM oxides and included the following stages: (1) $H_2SO_4$ leaching; (2) selective separation of REM as sulfate salt from Ni/Co sulfate solution; (3) metathesis purification reaction process for the conversion REM sulfate to REM carbonate; and (4) recovery of REM oxide from REM carbonate by heat treatment. Through $H_2SO_4$ leaching optimization, almost all the metal from the electrode active material of waste NiMH batteries was leached out. From the filtered leach liquor managing pH (at pH 1.8) with 10 M NaOH, REM was precipitated as hydrated $NaREE(SO_4)_2 \cdot H_2O$, which was then further valorized through the metathesis reaction process. From $NaREE(SO_4)_2 \cdot H_2O$ through carbocation, REM was purified as hydrated $(REM)_2CO_3 \cdot H_2O$ precipitate. From $(REM)_2CO_3 \cdot H_2O$ through calcination at 800 °C, $(REM)_2O_3$ could be recovered.

**Keywords:** spent NiMH batteries; circular economy; rare earths; valorization

## 1. Introduction

Owing to the strategic, economic, and industrial importance of rare earth metals (REMs), the EU has classified them as critical raw materials (CRMs). A criticality assessment of metals based on economic importance and supply risk done in 2017 resulted in the EU considering heavy rare earth metals (HREMs), light rare earth metals (LREM), and Sc as CRMs [1,2]. The EU, along with the US Department of Energy (DOE) [3] and the American Physical Society (APS), [4] have reported REM as critical for energy and emerging technologies. The global supply chain of critical metals like HREMs, LREMs, and Sc is mainly dominated by China, which has a market share of up to 95% [2]. The global distribution of primary resources for these critical REMs is quite uneven and currently dominated by China. Hence, the recovery of these CRMs from secondary resources is very important in order to secure these metals for industrial economies like Korea, where natural resources are scarce. The nickel metal hydride (NiMH) battery could be a secondary resource for REMs like Ce, La, and Nd, as it contains a significant amount of these elements (which are all REMs). By efficiently recycling NiMH waste batteries, the REMs can be recovered and circulated in the industrial ecosystem. This could simultaneously address issues like urban mining, environmental directive, and CRM supply chain challenges, and could secure the supply of these metals for industrial development.

With the emergence of more efficient batteries like the lithium-ion battery, the use of NiMH batteries has gradually decreased, which is leading to the NiMH batteries reaching end of life (EOL). Different industries around the world are recycling various batteries using either pyrometallurgy or hydrometallurgy processes, mainly targeting base metals like Co and Ni. Table 1 summarizes battery industries worldwide who are treating or recycling NiMH batteries in particular, and all types of batteries in general [5,6]. Though an industrial recycling process is available for NiMH batteries, the industrial-scale recovery of REMs has hardly been available. Numerous authors have reported on NiMH battery recycling research by hydrometallurgy, pyrometallurgy, or a hybrid of both [7–16]. Xia et al. reported on REM recovery by leaching–solvent extraction route where battery powder was leached by $H_2SO_4$, and REMs were then purified by solvent extraction using Cyanex 923. REM oxide was recovered using oxalic acid [16]. Yang et al. reported a HCl leaching and oxalic acid precipitation route for the recovery of REM oxide by calcination [11]. Korkmaz et al. have reported $H_2SO_4 + SO_2$ mixing, drying at 110 °C, roasting 850 °C and water leaching routes for the recovery of REMs from NiMH anode active materials [17]. Korkmaz et al. also reported a HCl acid leaching and oxalic acid precipitation route for the recovery of REM from waste NiMH batteries [18]. Innocenzi and Vegliò reported REM recovery by $H_2SO_4$ leaching followed by a $(REM)_2SO_4$ precipitation route using NaOH [19]. Though there have been several reports on the recovery of REMs by leaching precipitation, the REM oxide recovery leaching–precipitation–metathesis reaction route has hardly been discussed in the literature. As the industrial-scale recovery of REMs from waste NiMH is the primary challenge, in our current research we developed an industrially feasible recycling process where the circular economy goal can be achieved.

**Table 1.** Summary of battery recycling processes by various companies around the world (data from [5,6]).

| Company | Battery Type | Process/Technology Used | Location |
|---|---|---|---|
| Retriev Technologies | All types | Pyrometallurgy Hydrometallurgy | Trail, BC, Canada; Baltimore, OH, USA |
| Salesco Systems | All types | Pyrometallurgy | Phoenix, AZ, USA |
| AERC | All types | Pyrometallurgy | Allentown PA, USA; Hayward CA, USA; West Melbourne FL, USA |
| Dowa | All types | Pyrometallurgy | Japan |
| Japan Recycle | All types | Pyrometallurgy | Osaka, Japan |
| Sony Corp. & Sumitomo Metals and Mining Co. | All types | Pyrometallurgy | Japan |
| XStrata | All types | Pyrometallurgy Electrowinning | Horne Que, Nikkelverk Norway; Sudbury, ON., Canada |
| Accurec | All types | Pyrometallurgy | Mulhiem, Grenada |
| DK | All types | Pyrometallurgy | Duisburg, Greece |
| AFE Group (Valdi) | All types | Pyrometallurgy | Zurich, Switzerland; Rogerville, France |
| Citron | All types | Pyrometallurgy | Zurich, Switzerland; Rogerville, France |
| Euro Dieuze/SARP | All types | Hydrometallurgy | Lorraine, France |
| SNAM | Cd, Ni, MH, Li | Pyrometallurgy | Saint Quentin Fallavier, France |
| IPGNA Ent. (Recupyl) | All types | Hydrometallurgy | Grenoble, France |
| Umicore | All types | Pyrometallurgy Hydrometallurgy Electrowinning | Hooboken, Belgium |

In our current investigation, we developed a simple yet industrially feasible and sustainable REM oxide recovery process. The complete process is depicted in a flowsheet in Figure 1. As shown in

the figure, the developed process comprises sequential $H_2SO_4$ acid leaching followed by selective $NaREE(SO_4)_2 \cdot H_2O$ precipitation and purification of $(REM)_2CO_3 \cdot H_2O$ by carbocation–mixed REM oxide isolation by calcination. As shown in Figure 1, the process follows the integration of physical dismantling and classification followed by the chemical metallurgy process. The physical separation process comprises the following mechanical steps: (1) residual stored energy discharge; (2) grinding and calcination; and (3) classification and access of the electrode material. This is followed by the chemical metallurgy process, which has the following stages: (1) $H_2SO_4$ leaching; (2) separation of the REMs as sulfate salt from the Ni/Co sulfate solution; (3) the metathesis synthesis reaction process for conversion of REM sulfate to REM carbonate; and (4) REM oxide recovery by heat treatment. The novelty of this process is described below.

i.  All battery recycling processes by various companies around the world that have been reviewed mostly follow pyrometallurgical or pyrometallurgical-dominated processes, in contrast to our developed process which is hydrometallurical.
ii.  It is a simple acid leaching and precipitation process for the recovery of REMs.
iii.  Recovered REM sulfate is value-added through a simple carbocation reaction.

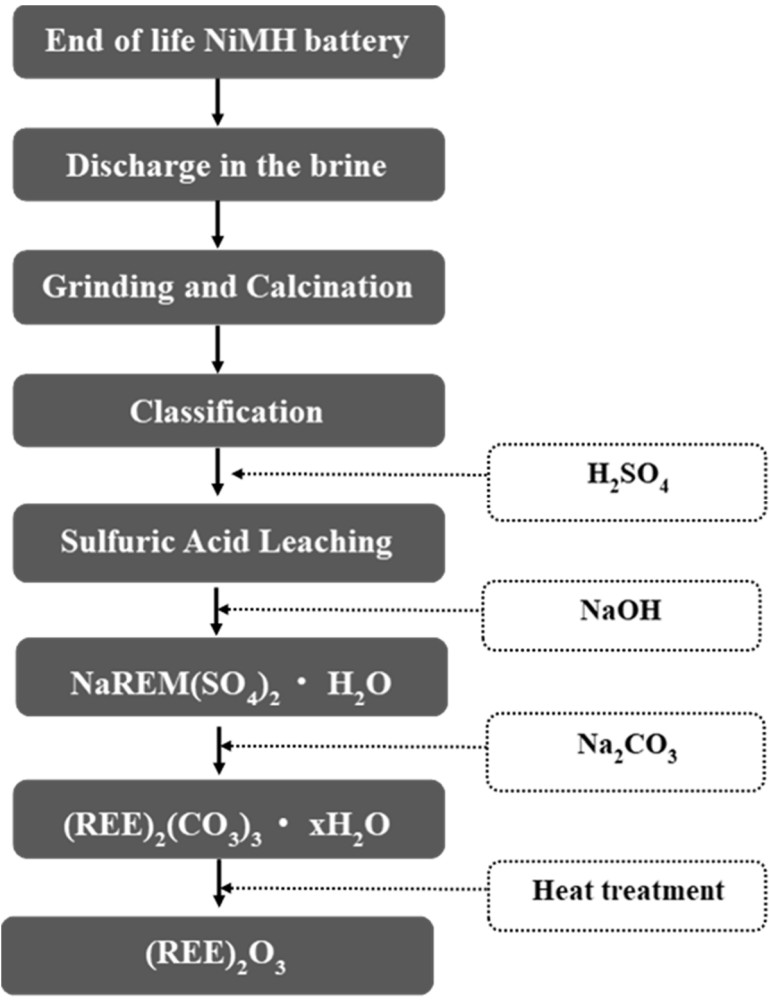

**Figure 1.** Experimental procedure for the recovery of rare earth oxide from a waste nickel metal hydride (NiMH) battery by a simple wet chemical valorization process. REM: rare earth metal.

## 2. Materials and Methods

### 2.1. Materials

Waste NiMH battery modules (Model: HHR-33AH72W6) were imported from Japan by GM-Tech Co., KOREA. After dismantling, electrode active material was supplied for the current investigation by GM-Tech Co., KOREA. All chemicals, $H_2SO_4$ (97%), NaOH (97%), and $Na_2CO_3$ (97%), were of analytical grade purchased from Daejung, Korea. The composition of recovered NiMH battery powder was analyzed by XRD (X-ray diffraction spectroscopy, XRD-6100, Shimadzu, Kyoto, Japan), XRF (X-ray fluorescence spectroscopy, ZSX Primus II, Rigaku, Houston, TX, USA) and MP-AES (Microwave Plasma-Atomic Emission Spectroscopy 4200, Agilent, Santa Clara, CA, USA). Thermal analysis was carried out by TGA (Thermo Gravimetric Analyzer, TG 8121, Houston, TX, USA).

### 2.2. Leaching and REM Separation

The leaching reactor used for leaching of the NiMH battery powder is shown in Figure 2. The reactor vessel was a three-necked round-bottom flask of 1 L capacity. The flask was jacketed for heating and controlling the reaction temperature. The reactor was equipped with an overhead agitator, and was driven by a variable-speed motor, a digital thermocouple to measure the temperature during continuous operation of the reactor, and a condenser for cooling. Before starting the leaching, the requisite volume of $H_2SO_4$ solution was added to the reaction flask and heated to the target temperature. Waste NiMH battery powder was added to the reaction vessel, which was then closed. After the reaction was complete, the leaching liquor was separated from residue through filtration. From the leach liquor, $NaREE(SO_4)_2 \cdot H_2O$ was separated by precipitation, controlling pH using NaOH while stirring at 400 rpm under pH control. Isolated $NaREE(SO_4)_2 \cdot H_2O$ was dried. For the metathesis reaction, the $NaREE(SO_4)_2 \cdot H_2O$ was added to the $Na_2CO_3$ solution and the reaction was allowed to complete. After reaction completion was achieved, the $(REM)_2CO_3$ was precipitated out and recovered through filtration. After drying, the $(REM)_2CO_3$ was calcined for $(REM)_2O_3$ recovery.

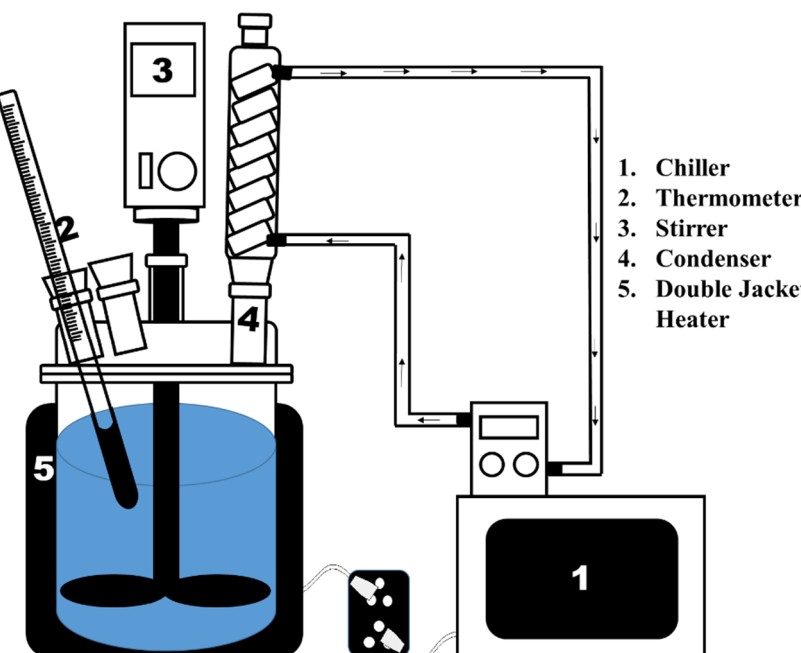

1. Chiller
2. Thermometer
3. Stirrer
4. Condenser
5. Double Jacket Heater

**Figure 2.** Schematic of the leaching reactor used for leaching waste electroactive powder material from a waste NiMH battery.

### 2.3. Characterization

Each of the isolated samples, including $NaREE(SO_4)_2 \cdot H_2O$, $(REM)_2CO_3$, and $(REM)_2O_3$, were analyzed by XRD (X-ray diffraction spectroscopy, XRD-6100, Shimadzu). The thermal decomposition behavior of the recovered sample was also analyzed by TGA (Thermo Gravimetric Analyzer, TG 8121, Rigaku).

## 3. Results and Discussion

### 3.1. Characterization of Waste NiMH Battery Material

Prior to characterization, the waste NiMH battery material module was discharged inside the brine. Grinding was followed by classification, and electroactive material of 100 mesh-size powder was recovered using a pulverizer. XRD (Figure 3) of the waste NiMH battery powder was analyzed. From XRD analysis, major phases such as $Ni(OH)_2$, NiO, Ni, $Ni_7Ce_2$, $Co_5Nd$, and $La_2O_3$ and minor phases such as NiO, NiOOH, $Ni_7Nd_2$, and $Co_3Nd$ were observed. XRF analysis for the same sample indicated Ni, Co, Ce, La, and Nd were the primary constituents in battery material with significant Ni content found in the waste NiMH battery. The composition of the recovered NiMH battery powder was also analyzed by MP-AES (Microwave Plasma Atomic Emission Spectroscopy 4200, Agilent). It was confirmed that Ni, Co, and REMs (La, Ce, Nd) comprised 45.8%, 8.5%, and 20.2% of the recovered battery powder, respectively (Table 2). As Table 2 indicates, the NiMH battery waste contained a significant amount of REMs (20.2%), which would be valuable CRMs for Korean industries.

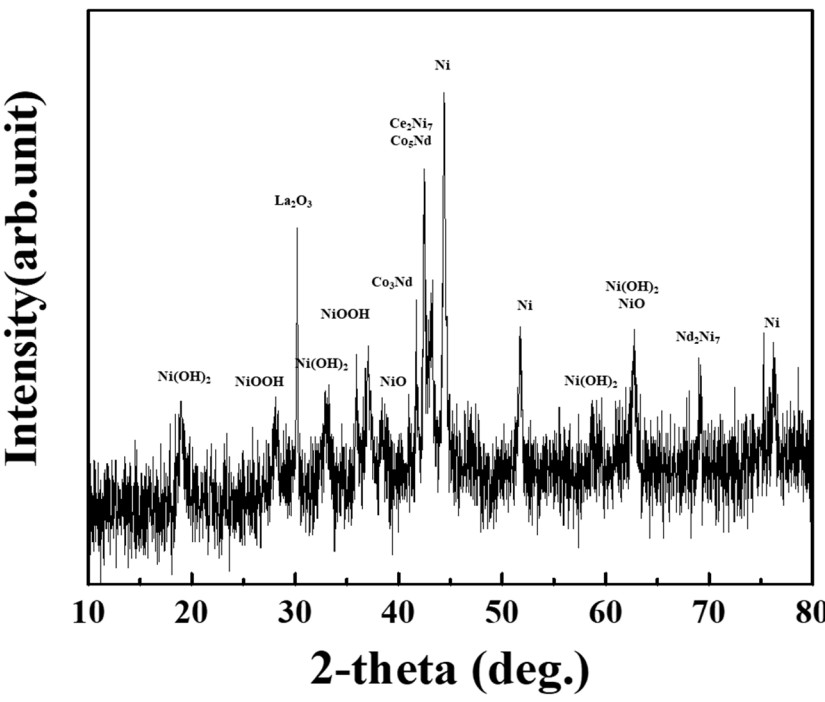

**Figure 3.** XRD pattern of ground waste NiMH battery powder.

**Table 2.** The composition (%w/w) of metal elements contained in the spent NiMH batteries.

| | Rare Earth Elements (REMs) | | | Total REMs | Ni | Co | Others |
|---|---|---|---|---|---|---|---|
| | **Ce** | **La** | **Nd** | | | | |
| Weight (%) | 10.4 | 6.7 | 3.1 | 20.2 | 45.8 | 8.5 | 25.5 |

### 3.2. Leaching Optimization of Waste NiMH Battery Material

For the development of a simple REM valorization process from the waste NiMH battery module, leaching parameters like temperature and reaction time were selected from the literature survey. Pietrelli et al. reported REM recovery from NiMH spent batteries using $H_2SO_4$ as a suitable lixiviant [20]. For the extraction of valuable REMs like Ce, La, and Nd from waste NiMH battery modules, important leaching parameters like the lixiviant ($H_2SO_4$) concentration and pulp densities were optimized, keeping the leaching temperature constant at 90 °C and leaching reaction time of 4 h. Temperature and time were considered from the reported studies by Pietrelli et al. [20]. The leaching optimization experiments were carried out and both parameters were varied at the same time. The lixiviant concentration ranged from 1 to 4 M of $H_2SO_4$, while pulp density ranged from 25 to 200 g/L. The reaction temperature was 90 °C the reaction time was 4 h. Figure 4a depicts the leaching behavior of Al, Co, Fe, Mn, Ni, and Zn, and Figure 4b depicts the leaching behavior of Ce, La, and Nd as a function of lixiviant concentration and pulp density. Figure 4a shows that Al, Co, Fe, Mn, Ni, and Zn extraction gradually increased as a function of lixiviant concentration and pulp density when both were abundant in the leaching process. Hence, higher efficiency, higher pulp density, and higher $H_2SO_4$ acid concentration could be suitable for efficient leaching process development. Similarly, as shown in Figure 4b, Ce, La, and Nd extraction increased up to 3 M $H_2SO_4$. Extraction decreased gradually when 4 M $H_2SO_4$ was used as a lixiviant. The same figure also indicates a higher amount of Ce, La, and Nd were extracted as function of pulp density. To understand the variable leaching phenomena observed in Figure 4, the leaching chemistry needed to be understood. As the focus of the investigation is only REMs, the possible leaching reaction chemistry is explained below. As XRD indicated, La, Ce, and Nd (REMs) presented mainly as REM oxide and a Co/Ni alloy compound. Therefore the leaching chemistry can be explained using Equations (1)–(3) presented below.

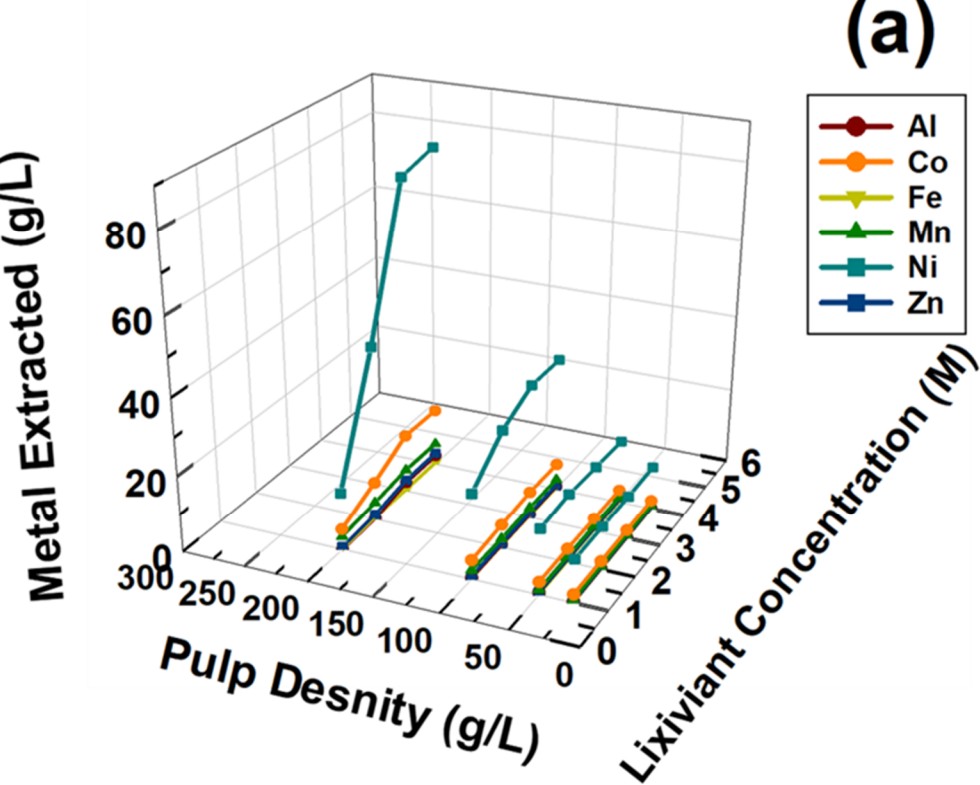

**Figure 4.** *Cont.*

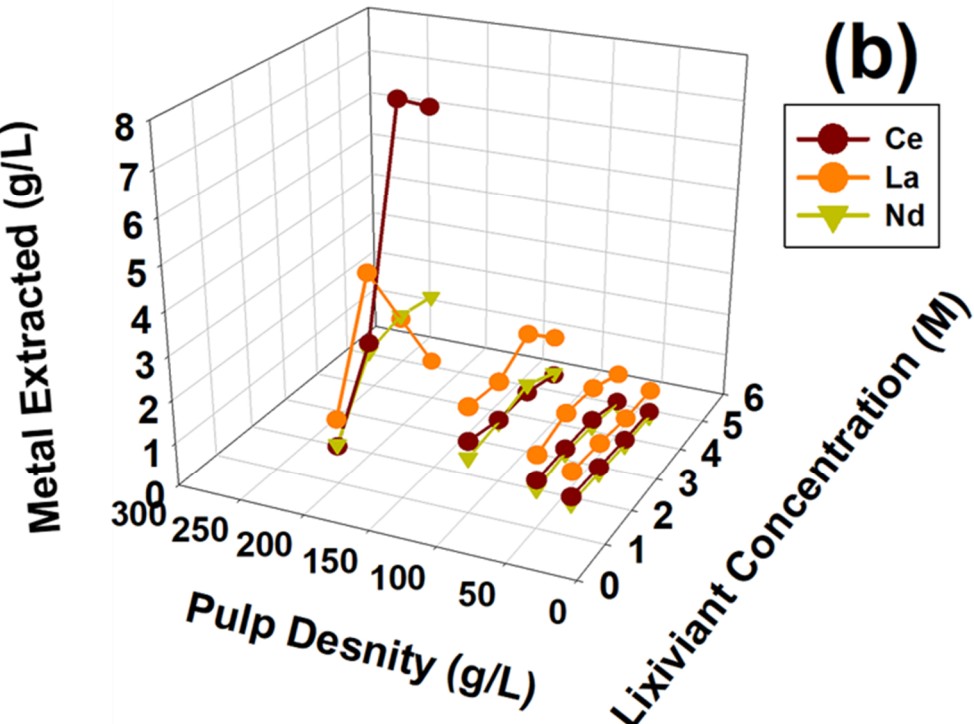

**Figure 4.** Effect of lixiviant concentration and pulp density on the extraction of metals (**a**) Al, Co, Fe, Mn, Ni, and Zn, (**b**) Ce, La, and Nd, from waste NiMH battery material.

$$La_2O_3 + 3H_2SO_4 \rightarrow La_2(SO_4)_{3(aq)} + 3H_2O \tag{1}$$

$$(REM)_2Ni_x + (x+2)H_2SO_4 \rightarrow (REE)_2(SO_4)_{3(aq)} + xNiSO_{4(aq)} + (x+2)H_3O^+ \tag{2}$$

$$Co_xNd + (x+3)H_2SO_4 \rightarrow Nd_2(SO_4)_{3(aq)} + xCoSO_{4(aq)} + (x+3)H_3O^+ \tag{3}$$

*3.3. Separation and Recovery of REM by Metathesis Reaction*

Though base metal extraction for Ni and Co was more efficient with higher pulp density and higher lixiviant concentration, REM extraction was not much more efficient at higher lixiviant concentrations. As the main focus of the investigation was REM recovery, conditions such as using 1 M of $H_2SO_4$ acid as a lixiviant and a pulp density of 25 g/L were considered for further studies. Sufficient volume of leach liquor was generated using 1 M $H_2SO_4$ acid at a pulp density of 25 g/L, at a temperature of 90 °C with a reaction time of 4 h. Followed by solid–liquid separation, REMs were separated the leach liquor through precipitation using NaOH. The initial pH of the leaching liquor was about 0.05 after solid–liquid separation. The pH of leach liquor was adjusted by slowly adding 10 M NaOH to the leach liquor under constant stirring. As the pH of the solution was increased slowly to 1.8, REMs were precipitated, leaving Co, Ni, and other metals in the leach liquor. The precipitated REMs were separated from the solution by simple filtration and the sample was washed several times to remove soluble components. After washing, the samples were dried overnight in an oven at 80 °C, and were then characterized by XRD, ICP-MS, and TGA.

The recovered REM powder was analyzed by XRD and the obtained XRD pattern is presented in Figure 5. The powder was identified as a mixture of $NaNd(SO_4)_2 \cdot H_2O$, $NaCe(SO_4)_2 \cdot H_2O$, and $NaLa(SO_4)_2 \cdot H_2O$, which corresponded to JCPDS # 40–1481, JCPDS # 86–0526, and JCPDS # 82–1199, respectively. From now on, the $NaNd(SO_4)_2 \cdot H_2O$, $NaCe(SO_4)_2 \cdot H_2O$, and $NaLa(SO_4)_2 \cdot H_2O$ mixture

will be termed ($NaREE(SO_4)_2 \cdot H_2O$). The content of Ce, La, and Nd impurities including Ni and Co was analyzed by MP-AES by dissolving the powder. This is presented in Table 3. The MP-AES confirmed that Ce (17.2%), La (13.1%), and Nd (5.4%) were the main metal components in the precipitate, whereas Co and Ni impurities only made up 0.01%. Hence, selective separation of $NaREE(SO_4)_2 \cdot H_2O$ using a precipitation technique by adding NaOH to the leachate is an efficient method to recover REMs from leach liquor.

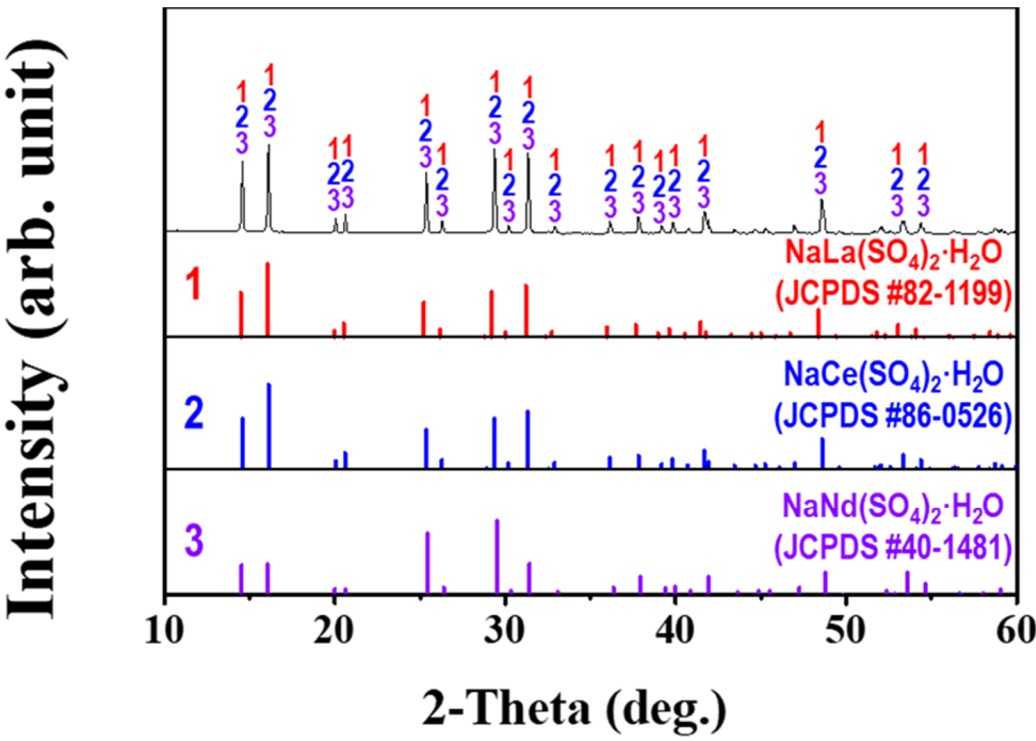

**Figure 5.** XRD pattern of isolated REM (the $NaNd(SO_4)_2 \cdot H_2O$, $NaCe(SO_4)_2 \cdot H_2O$, and $NaLa(SO_4)_2 \cdot H_2O$ mixture is referred to as $NaREE(SO_4)_2 \cdot H_2O$)) precipitates.

**Table 3.** The composition (%w/w) of the metal elements contained in $NaREE(SO_4)_2 \cdot H_2O$.

| # | Rare Earth Elements | | | Ni + Co |
|---|---|---|---|---|
| | Ce | La | Nd | |
| Weight (%) | 17.2 | 13.1 | 5.44 | 0.01 |

From the isolated $NaREE(SO_4)_2 \cdot H_2O$, a carbonation reaction using $Na_2CO_3$ was completed, and the REMs were purified as anhydrous $(REE)_2(CO_3)_3 \cdot xH_2O$. The carbonation reaction was carried out by adding the isolated $NaREE(SO_4)_2 \cdot H_2O$ powder to 200 mL of $Na_2CO_3$ solution in 500 mL of reactor, and the reaction was carried out for 5 h. The stoichiometric ratio of 1:1.1 for REM(III) with $CO_3^{2-}$ was maintained for the purification of anhydrous $(REE)_2(CO_3)_3 \cdot xH_2O$. The carbonation reaction was carried out in two different temperatures (i.e., room temperature (RT) and at 70 °C). After the reaction was completed, the isolated sample was dried and analyzed by XRD. The XRD pattern is depicted in Figure 6. The XRD pattern in Figure 6 indicates that there was almost no difference in XRD patterns between the two different samples isolated after the reaction at both room temperature and at 70 °C. The XRD pattern was confirmed to be $(CeLa)_2(CO_3)_3 \cdot 4H_2O$, $(LaNd)_2(CO_3)_3 \cdot 8H_2O$, and $NaNd(CO_3)_2 \cdot 6H_2O$, which correspond to JCPDS #06–0076, #30–0678, and #30–1223, respectively.

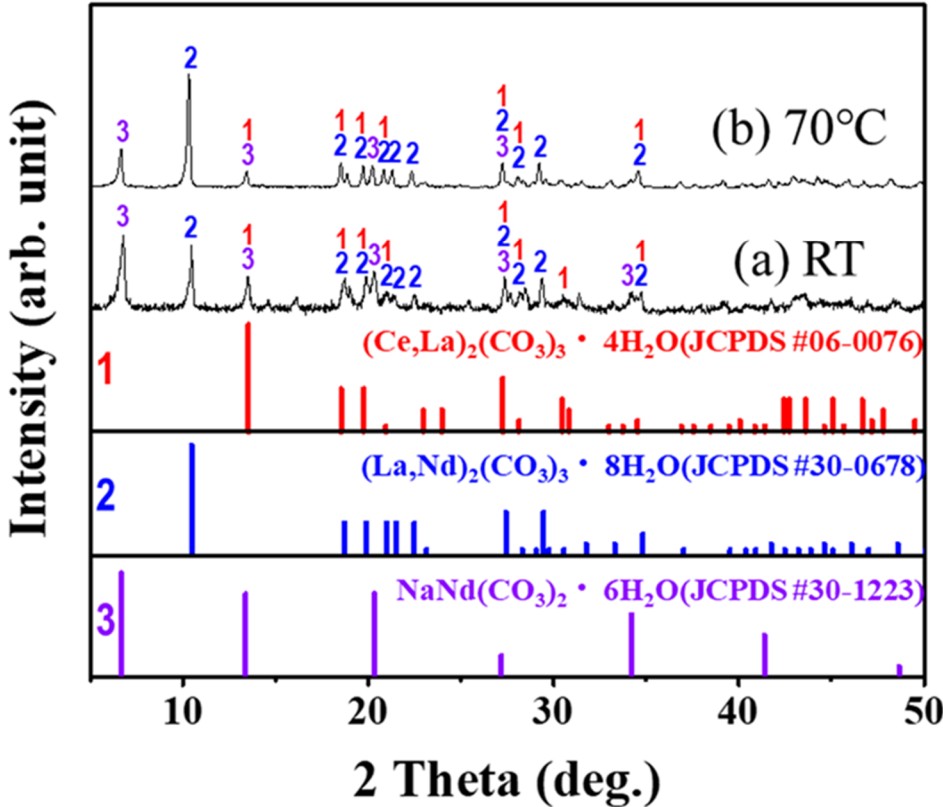

**Figure 6.** XRD pattern of purified $(REM)_2CO_3 \cdot H_2O$ powder, obtained through a metathesis reaction from $NaREE(SO_4)_2 \cdot H_2O$ precipitates.

The XRD pattern observed is a mixture of all three REM carbonates, so it will be presented as $(REM)_2CO_3$. The $(REM)_2CO_3$ was analyzed by TGA (Figure 7). The TGA figure indicates two stages of weight loss as temperature increased to 1000 °C. In the first step, the weight loss was 26.1% between room temperature and 250 °C. The weight loss is associated with loss of water and conversion of hydrated $(REM)_2CO_3 \cdot H_2O$ to anhydrous $(REM)_2CO_3$. In the second step, the weight loss was 44.7% from room temperature to 800 °C. At 800 °C, $(REM)_2CO_3$ was converted to $(REM)_2O_3$ by loss of $CO_2$. To understand the $(REM)_2O_3$ formation through the metathesis reaction, the $(REM)_2CO_3 \cdot H_2O$ sample was calcined at 800 °C for 1 h and characterized by XRD analysis. The XRD pattern is represented in Figure 8. The XRD peak in the XRD analysis indicates that the sample was a mixture of three types of rare earth oxides (i.e., $Nd_{0.5}Ce_{0.5}O_{1.75}$, $La_2O_3$, and $Nd_6O_{11}$). Hence, pure $(REM)_2O_3$ can be valorized through heat treatment of $(REM)_2CO_3$ at 800 °C.

A complete flowsheet describing the recovery of $(REM)_2O_3$ from a waste NiMH battery via a simple wet chemical valorization process was developed and is shown in Figure 9. The figure depicts the sequential process, i.e., $H_2SO_4$ acid leaching, $NaREE(SO_4)_2 \cdot H_2O$ precipitation, and metathesis reaction for the complete recycling of waste NiMH batteries. The process was completed with the integration of physical separation followed by the chemical metallurgy process. The physical separation process was followed by residual stored energy discharge, grinding and calcination, classification, and access to the electrode active material. In the chemical metallurgy process, the electrode active powder material was extracted by leaching, REMs were separated selectively by precipitation, and purified by carbocation and calcination processes. The chemical metallurgy process sequentially followed (i) 1 M $H_2SO_4$ leaching from a pulp density of 25 g/L at 90 °C, (ii) selective separation of REMs as $NaREE(SO_4)_2 \cdot H_2O$ salt leaving Ni/Co sulfate solution, (iii) conversion of $NaREE(SO_4)_2 \cdot H_2O$ to $(REM)_2CO_3 \cdot H_2O$ through carbocation, and (iv) synthesis of REM oxide from $(REM)_2CO_3 \cdot H_2O$ by calcination at 800 °C. The mixed REM oxide could be further purified to individual REMs by further

investigations, mainly by hydrometallurgy. As this is a complete systematic sequential process for the recovery of mixed critical REMs like Ce, La, and Nd, it could address the circular economy and recycling challenges associated with waste NiMH batteries.

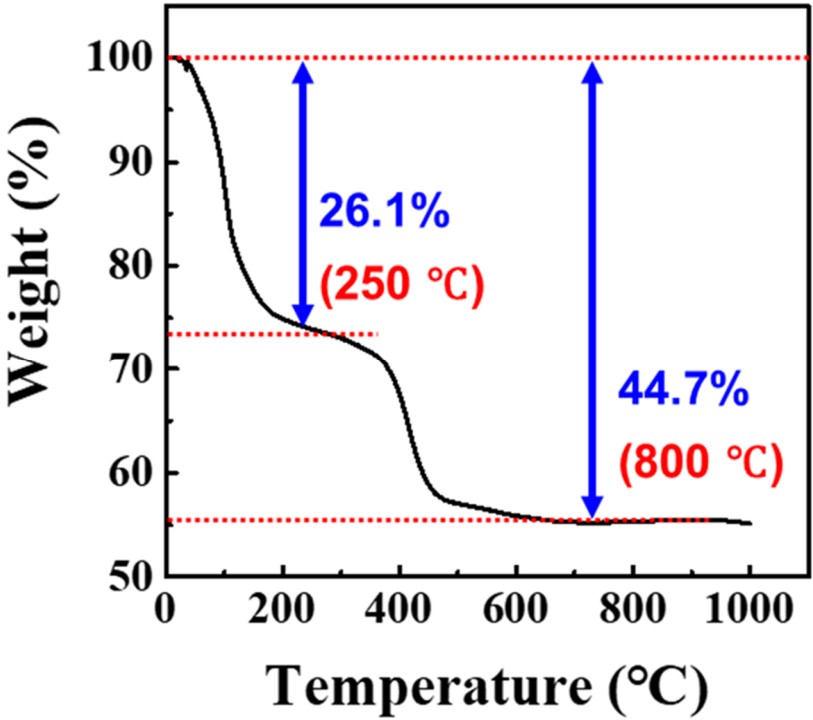

**Figure 7.** Thermal characteristics of purified (REM)$_2$CO$_3$·H$_2$O powder.

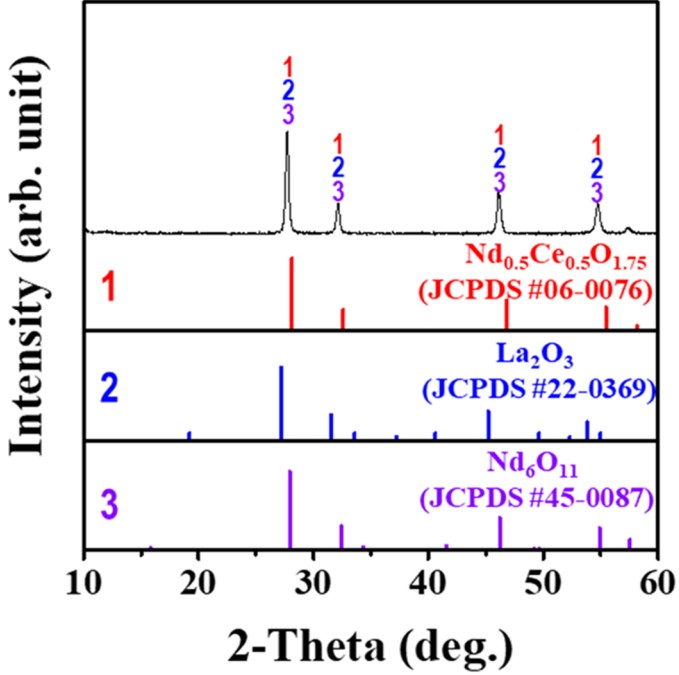

**Figure 8.** XRD pattern of pure (REM)$_2$O$_3$ powder, obtained through the calcination of (REM)$_2$CO$_3$·H$_2$O powder.

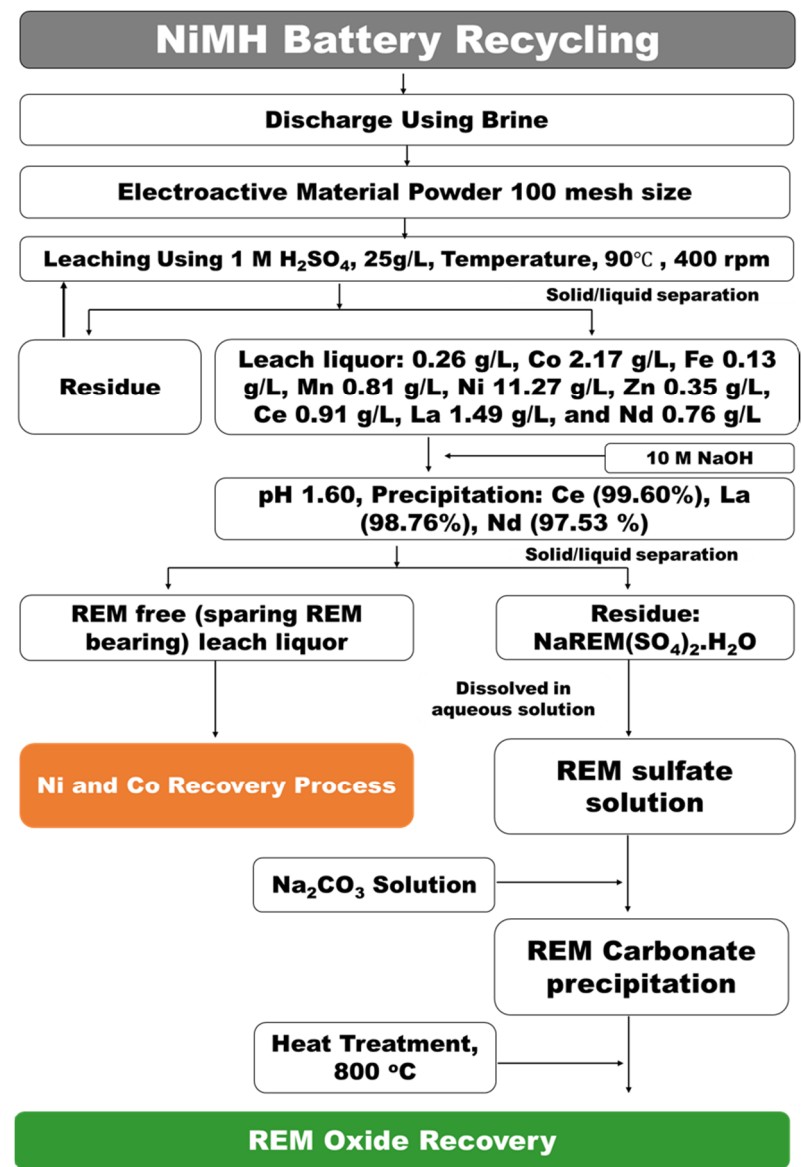

**Figure 9.** Process flow sheet for the recovery of rare earth metal oxide from waste NiMH batteries by a simple wet chemical valorization process.

## 4. Conclusions

Through the $H_2SO_4$ leaching–selective precipitation–metathesis reaction route of waste NiMH batteries, REM oxide could be recovered quantitatively. Through $H_2SO_4$ leaching optimization, almost all metal from the electrode active material of waste NiMH batteries was leached out. From the filtered leach liquor with pH managed (pH 1.8) by 10 M NaOH, REMs were precipitated as hydrated $NaREE(SO_4)_2 \cdot H_2O$, which was further valorized through a metathesis reaction process. Isolated $NaREE(SO_4)_2 \cdot H_2O$ was further purified through a carbocation reaction using $Na_2CO_3$. From $NaREE(SO_4)_2 \cdot H_2O$ through carbocation, REMs were purified as hydrated $(REM)_2CO_3 \cdot H_2O$ precipitate. From $(REM)_2CO_3 \cdot H_2O$ through the calcination process at 800 °C, $(REM)_2O_3$ could be recovered. The mixed $(REM)_2O_3$ could be further purified through hydrometallurgy, such as dissolution followed by solvent extraction purification. The process is very simple, versatile, and easy to implement in the industry. Advantages of the process are that it is simple, versatile, and flexible, and industrial mass production is feasible. It is also sustainable and eco-efficient, and produces minimal

waste emissions. The process addresses several issues concurrently, including waste valorization, environment management, critical REM circular economy, and urban mining.

**Author Contributions:** N.-K.A. and B.S. conceived and designed the experiments; H.-W.S. managed project and analyzed data, and D.-W.K. contributed for funding arrangement, writing and editing the manuscript. B.S. prepared draft and edited the manuscript.

**Funding:** This work was supported by the Korea Evaluation Institute of Industrial Technology (KEIT), which is funded by the Ministry of Trade, Industry and Energy, Republic of Korea (Project No. 10077752).

**Conflicts of Interest:** The authors declare no conflict of interest.

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
