# Peer review of "Recovery of Rare Earth Oxide from Waste NiMH Batteries by Simple Wet Chemical Valorization Process"

_metals, doi:10.3390/met9111151_

Round 1

Reviewer 1 Report

The article is interesting and offers useful experimental data for an ecological hydrometallurgical process for recovering rare metals from used NiMH batteries.

Author Response

Comment:

The article is interesting and offers useful experimental data for an ecological hydrometallurgical process for recovering rare metals from used NiMH batteries.

Response:

Thanks to the reviewer for putting lots of interest and time.

Reviewer 2 Report

The authors tried to develop a technique for recovering the REE from the waste NiMH battery using combined leaching and calcination. Overall the technique and approach are okay. There are however a number of things that need to be addressed as mentioned below and in the attached comments. 

There are numerous typographical and grammatical errors. The authors need to check the manuscript line by line for errors. Some examples of errors can be seen in the attached file.  The authors need to explain in more details the basis of the selection of the temperature and time of experiments / leaching in the section 2. The authors need to supply the XRD pattern of the original waste battery materials Comments on errors, standard deviation, repeatability and number of experiments for each data point will need to be incorporated in the manuscript Figure 3(a) must be on top of Figure 3(b) Correct the axis title of Figures 3(a) and 3(b). Desnity --> Density Some image (e.g.SEM) of the REM2O3 powder obtained from the process can be incorporated in the paper.  What is the composition of the REM2O3, i.e. La, Ce, Nd, O? What is the purity of the final oxide Need to briefly explain what would be the possible further ways to separate La, Ce, Nd from the oxide.  Add reaction during calcination in the manuscript

Author Response

Comment

There are numerous typographical and grammatical errors. The authors need to check the manuscript line by line for errors. Some examples of errors can be seen in the attached file. 

Response:

Comment

The authors need to explain in more details the basis of the selection of the temperature and time of experiments / leaching in the section 2.

Response:

May be reviewer has over looked. I am sincerely request the reviewer please see the section 3.2, page 5 and line 133-135.

Comment

The authors need to supply the XRD pattern of the original waste battery materials.

Response:

We have supplied the instructed information as Figure 3, subsequently all figure number also corrected.

Comment

 Comments on errors, standard deviation, repeatability and number of experiments for each data point will need to be incorporated in the manuscript Figure 3(a) must be on top of Figure 3(b). Correct the axis title of Figures 3(a) and 3(b).

Response:

We have placed the figure as advised. Now the figure is changed to Figure 4a and Figure 4b.

Comment

Desnity --> Density Some image (e.g.SEM) of the REM2O3 powder obtained from the process can be incorporated in the paper. 

Response:

As the process speaks about the recovery process, SEM hardly serves the purpose. Again manuscript already has 9 figures.

Comment

What is the composition of the REM2O3, i.e. La, Ce, Nd, O? What is the purity of the final oxide Need to briefly explain what would be the possible further ways to separate La, Ce, Nd from the oxide.  Add reaction during calcination in the manuscript 

Response:

The composition of the obtained powder is given. The reviewer perhaps looking for the stoichiometry of REM2O3, which needs further investigation, which is not the scope of the current investigation. Yes, it is possible to separate La, Ce, and Nd, but this itself self-standing problem needs precise investigation, but the discussion would be speculative, hence, we are avoiding. Still, we incorporated a statement, in conclusion, page 13, line 248-250.   

NOTE: All correction suggested in the  attached file 

Reviewer 3 Report

The paper studies the recovery of a mixture of rare earth oxides contained in NiMH batteries. The process is carried out through physical (dismantling, crushing ...) and chemical (leaching, selective precipitation and calcination) stages.

The paper is very interesting, it is well written, the experimental methods are quite well described (can be improved), the discussion of the results is well done and the conclusions are supported by the results obtained, although at the scale at which they have been performed the experiments (laboratory scale, with very small amount of sample) no conclusions can be drawn about the viability of the process at the industrial level.

In my opinion, the manuscript is of interest to the readers of the Metals and could be included in the Special Issue Metal Removal and Recycling.

However, I raise some questions to which the authors must answer:

a) The authors should emphasize the innovative aspects of the process they have studied, compared to the background they have presented in Table 1.
b) The dismantling stage of the batteries must be much better explained. I suggest an new experimental section 2.2. Dismantling. To explain this section well is essential to understand the manuscript well. What is a grinding and calcination stage for? How is the isolation of the cells carried out? and its opening ?. Authors should explain this stage more extensively.
c) Figure 2 shows the position of the impeller

Consequently, the manuscript can be published with minor corrections

Author Response

Comment
a) The authors should emphasize the innovative aspects of the process they have studied, compared to the background they have presented in Table 1.

Response:

We have added a novelty statement on page 3 and line 77-81.

Comment

b) The dismantling stage of the batteries must be much better explained. I suggest a new experimental section 2.2. Dismantling. To explain this section well is essential to understand the manuscript well. What is a grinding and calcination stage for? How is the isolation of the cells carried out? and its opening ?. The authors should explain this stage more extensively.

Response:

Dismantling is the scope of the study, hence, we have not included here.

Comment

c) Figure 2 shows the position of the impeller

Response:

We have corrected it as directed.